# WSN-SES/MB: System Entity Structure and Model Base Framework for Large-Scale Wireless Sensor Networks

**DOI:** 10.3390/s21020430

**Published:** 2021-01-09

**Authors:** Su Man Nam, Hyung Jong Kim

**Affiliations:** 1DUDU Information Technologies, Inc., Seoul 08501, Korea; sumannam@gmail.com; 2Department of Information Security, Seoul Women’s University, Seoul 01797, Korea

**Keywords:** wireless sensor networks, network simulator, system entity structure, model base, discrete event system specification

## Abstract

Large-scale wireless sensor networks are characterized by stringent energy and computation restrictions. It is exceedingly difficult to change a sensor network’s environment configurations, such as the number of sensor nodes, after deployment of the nodes. Although several simulators are able to variously construct simulation models for sensor networks before their deployment, the configurations should be modified with extra human effort as the simulators cannot freely generate diverse models. In this paper, we propose a novel framework, called a system entity structure and model base for large-scale wireless sensor networks (WSN-SES/MB), which is based on discrete event system specification formalism. Our proposed framework synthesizes the structure and models for sensor networks through our modeling construction process. The proposed framework achieves time and cost savings in constructing discrete event simulation-based models. In addition, the framework increases the diversity of simulation models by the process’s pruning algorithm. The simulation results validate that the proposed framework provides up to 8% time savings and up to 23% cost savings as compared to the manual extra effort.

## 1. Introduction

Wireless sensor networks (WSNs) have been widely employed in area monitoring, military applications, and fire detection applications [1,2,3,4,5,6,7]. The sensor networks record the conditions of environments to monitor the applications and organize the collected data at a central location. Figure 1 shows a WSN example, comprised of numerous sensor nodes for recording the conditions and a single base station (BS) for collecting the data. When a sensor node detects a real event (e.g., the appearance of a tank), the node generates an event report. It forwards the report via multiple hops toward the base station. Although the sensor network automatically operates the collection and analysis of the data without human intervention, the network has profound impact on its lifetime, since it is affected by tiny configurations (e.g., the field size, the BS location etc.) after deployment of the sensors [8,9,10].

In a sensor network, the configuration of the network environment is very important because it is exceedingly difficult to change [2,11]. In addition, it consumes a lot of resources to add or modify the network configuration (a sensor node addition, its routing path modification, etc.) one installed [8,9,10,12]. The lifetime of the network is heavily determined by the selection with respect to energy consumption, node deployment, scalability, connectivity, coverage, and security [5]. Therefore, before deploying sensors in the field, we should utilize some techniques, such as modeling and simulation [13,14,15], to predict the behavior and performance of a network with the consideration of the network environment [8,16,17].

Researchers have proposed several simulators to predict the behavior and performance of WSNs [16,17,18,19]. Even though these simulators accurately model sensors and simulate the models under various operational scenarios, they may place some restrictions on the model construction for the sensor networks. It is necessary to include the synthesis technology based on a process without any extra human effort (i.e., hardcoding). Thus, the WSN simulators should be able to freely manipulate the network configuration, to model and simulate the behavior of the sensors, and to evaluate the factors of the nodes.

To this end, the system entity structure and model base (SES/MB) framework [13,15,20,21,22,23,24] synthesizes entity structures and behavioral characteristics to generate diverse simulation models based on a transformation process. The SES systematically expresses the structures of complex systems and the MB represents behavioral models of the systems using Discrete Event System Specification (DEVS) formalism [13,14,25,26]. Although the SES/MB uses the structural expression and the atomic DEVS models to build the executable simulation models, it is difficult to synthesize the structure and the model including the configuration parameters with little time and cost. For example, to configurate or modify a BS’s location, an atomic model of the BS should be manually hardcoded with the location parameters (X and Y coordinates). For this reason, there are limits on synthesizing the entity structures and behavioral models, including the environment parameters. It is necessary to achieve a new SES/MB framework to effectively synthesize them for WSNs.

In this paper, we propose the SES/MB for WSNs (hereafter called the WSN-SES/MB) framework to automatically synthesizes the structure and models through our modeling construction process. The WSN-SES/MB uses the complicated synthesizing process of the WSN modeling construction (i.e., transformation process) to achieve time and cost savings. In our proposed framework, the transformation processes the entity structures and DEVS behavioral models of a sensor network with the parameters of the network environment. The transformation process expresses the structure of the network as an SES tree, selects the specific entities and the environment parameters using our pruning algorithm, and coverts the SES into a pruned entity structure (PES) tree. This process constructs a hierarchical and structural simulation model by combining the tree structure, the parameters, and the atomic DEVS models. Experimental results indicate that our framework improves the execution time by up to 8% and the central processing unit (CPU) utilization cost by up to 23% as compared to the manual model synthesis.

The main contributions of this paper are as follows:Applying the SES/MB for synthesizing the structure and models and automating the complicated synthesizing process of WSN modeling construction;Saving time and cost in synthesizing various models by proposing a novel transformation process;Achieving efficiency of the WSN-SES/MB framework with a new pruning algorithm in our transformation process;Increasing diversity of synthesized simulation models by introducing the WSN-SES/MB framework;Facilitating the modeling and simulation of large-scale WSNs with all of the above.

The rest of this paper is organized as follows: Section 2 introduces the existing general simulators for sensor networks. Section 3 presents the SES/MB framework for synthesizing the structure and the models. We offer a detailed description of the WSN-SES/MB framework in Section 4. In Section 5, we present a performance evaluation of the WSN-SES/MB framework using analysis and simulation. We draw conclusions at the end of this paper.

## 2. WSN Simulators

Many researchers have proposed several simulators to predict the behavior and performance of WSNs. There are the Network Simulator-3 (NS-3) [18], the Objective Modular Network Testbed in C++ (OMNET++) [19], TinyOS simulator (TOSSIM) [16], probabilistic wireless network simulator (Prowler) [27,28], simple NEST application simulator (Siesta) [29], Ashut [30], routing modeling application simulation environment (Rmase) [30], a sensor, environment and network simulator (SENS) [31], Fine-Grained Sensor Network Simulator (ATEMU) [32] and Avrora [17]. NS-3 and OMNET ++ use the IEEE 802.15.4 standard [33] to evaluate protocols and algorithms based on a general-purpose network simulator. The other simulators are widely used as hybrid simulations that connect directly to physical sensors on a hardware platform. In addition, the recently proposed CupCarbon [34,35] is a new platform for designing and simulating a smart city and Internet of things based on WSNs.

NS-3 supplements the disadvantages of unnecessary coupling, complex structure, and low flexibility from NS-2 [36]. The NS-3 provides a user-friendly environment for choosing between C++ and Python languages. In addition, this simulator is an open-source network simulator that operates in a discrete, event-based simulation. The simulator uses various modules that modeled network topology and communication methods such as Zigbee, WiFi, and LTE. The simulator monitors network traffic using several tools such as FlowMonitor [37] and Wireshark [38]. Although the NS-3 is widely used for network simulations, it is not enough to study specific libraries (routing protocols, security protocols etc.) for a sensor network compared to regular network simulators [39].

OMNET++ is a modular discrete event network simulation framework used primarily for building network simulators. The OMNeT++ offers a low complexity compared to the NS-2 and the NS-3 by mandating the communication between modules using predefined connections. This framework is regarded as an extensible, modular and component-based C++ simulation library framework for building wireless network simulations. Basically, the OMNeT++ is not a simulator but it provides frameworks and tools for writing simulation scenarios. The OMNeT++ is not enough to perform simulations with large sensor nodes like the NS-3.

TOSSIM, Prowler, Siesta, and Rmase were developed in the network embedded software technology (NEST) project. The TOSSIM verifies the behavior of TinyOS and applications on sensor nodes, and Prowler evaluates the algorithms of wireless networks. Siesta is a middleware simulator that validates NEST applications and middleware functionality, and Rmase evaluates network topology and routing protocol functionality based on Prowler. TinyOS Scalable Simulation Framework (TOSSF), PowerTOSSIM, and Tython have been proposed to resolve these problems, but still depend on specific software [40,41].

SENS provides a simulation environment that organizes applications, network communications, and the physical environment into modules. This simulator also depends on specific libraries. ATEMU simulates sequentially at the system clock interval of the sensor, while Avrora runs one thread per node; those threads synchronize only when necessary. Although these simulators can predict resource usage, it is difficult to simulate a large number of sensors due to the long simulation time according to the high simulation precision.

CupCarbon is a Smart City and Internet of Things Wireless Sensor Network (SCI-WSN) simulator based on multi-agent and discrete event simulation. This simulator can model and simulate sensor networks on a digital geographic interface of OpenStreetMap. The simulator’s objective is to design, visualize, debug, and validate distributed algorithms for monitoring, environmental data collection, etc. In addition, the simulator creates environmental scenarios such as fires, gas, mobiles. It supports scientists in testing their wireless topologies and protocols. Although the CupCarbon can simulate and monitor various situations based on geolocation, it is difficult to use various routing protocols [42,43] of the sensor network.

Although such simulators are being used due to their characteristics, it is difficult to model using the implicit simulation and modular aspects of each component of the sensor node [44,45]. In addition, it is difficult to observe the behavior of the sensors under various environmental conditions. To solve these problems, research for modeling and simulating the sensor network based on the DEVS formalism is underway. Cell-DEVS [45,46,47] is the DEVS-based formalism that defines spatial models as a cell space. DEVS-C++ [48] is also based on the DEVS formalism, and is used for performance measurement of large-scale sensor networks in [9,10,49]. Despite these DEVS engines providing various model implementations and model reuse for the sensor network, they have limitations when synthesizing the structure and the DEVS models without any extra human effort.

Table 1 shows such a WSN simulator comparison to express the effectiveness of our proposal.

## 3. SES/MB Framework

The SES/MB framework generates simulation models of modular and hierarchical systems through the SES ontology and the classical workflow of modeling [20]. The SES/MB framework is used for the multifaceted modeling simulation of a standalone system. The framework uses a transformation process to synthesize entity structures and DEVS models to generate a simulation model, as follows:(1)SES→Pruning→PES→Composition Tree+MB→Simulation Model

In Equation (1), the SES is represented by a tree structure containing alternative edges starting at decision nodes. The SES has to be pruned to select a specific design through an algorithm and to build a specific model variant. This pruning process constructs a Pruned Entity Structure (PES) that defines one model variant. A composition tree derived from the PES includes all the necessary information to build a modular and hierarchical simulation model using predefined basic components from the MB.

Figure 2 shows the main transformation process in the SES/MB framework. A simulation model is built through trees in the SES and PES bases (Figure 2a,b) and basic models in a model base (Figure 2c) according to the process. The SES trees portray the structural knowledge of a system and the model base stores the real models that represent the behavioral knowledge of the system. In this Section, the SES and PES bases are discussed in Section 3.1, and the model base is introduced in Section 3.2.

### 3.1. SES and PES Bases

The SES is an ontology-based framework for specifying a set of modular, hierarchical system designs by a single formal description [50]. A system design consists of a specific system structure and a set of system parameters. That means that the SES does not define any dynamic behavior on its own. The SES is represented by a tree structure comprising entity nodes, descriptive nodes and attributes [51]. Different system structures can be coded in the SES trees. Modeling and simulation entity nodes are linked to basic models organized in MB. The attributes of an entity node correspond to the parameters of the associated basic model.

As shown in Figure 2a, an SES tree is represented as a labeled tree with attached attributes, which satisfies the following axioms: uniformity, strict hierarchy, alternating mode, valid brother, and attached variables [22]. There are three types of node in the tree. The entity node (like ABC in Figure 2a) consists of a composite entity and an atomic entity, which represents a real-world object, denoted by |. The composite entity connects with other entities (composite or atomic entity); while the atomic entity is located at the end of the tree. The composite entity is attached with variables, which have several aspects and/or specializations. An aspect node (like abc-dec in Figure 2a) is connected by a single vertical line from an entity. This node represents one decomposition of the entity. Each node includes coupling information. In addition, an aspect node is able to be a multiple-decomposition (like cs-mdec in Figure 2a) of another entity, denoted by ⦀. The multiple-decomposition is used to represent the number of children of the entity with coupling information. The coupling information can be specified as properties of descriptive nodes of the type decomposition or multi-decomposition, and consist of pairs of entity names and port names, such as (source entity, source port, sink entity, sink port). The specialization node (like ab-spec in Figure 2a) is connected by a double vertical line to an entity, denoted by ‖. This node describes the taxonomy of the entity. The node represents the way in which a specialized entity is chosen in the pruning process according to selection rules. In [15], The SES includes three tuples as follows
(2)SES=〈Item, Relation, Attibute〉
where

Item=E∪A∪S: Set of items:E: set of entities;A: set of aspects;S: set of specialization;

Relation=Asp∪Spec: set of relations among items:$Asp\subset E\times A\times 2^{E}$: aspect relationships;$Spec\subset E\times S\times 2^{E}$: specialization relationship;

Attributes=Coup∪Rules: set of attributes attached to items:$Coup:\;A\to 2^{\left(E.IO\times E.IO\right)}$: couplings attached to aspects;$Rules:\;S\to 2^{R}$: selection rules attached to specializations and multiple-decomposition.

In Equation (2), IO represents a set of input and/or output ports of respective entities, R is a set of selection rules in the form of (cond→E). According to Equation (2), the SES tree of Figure 2a is defined as follows:*E* = {ABC, AB, CS, A, B, C};*A* = {A-dec, cs-mdec};*S* = {ab-spec};*Asp* = {(ABC, abc-dec, {AB, CS}), (CS, cs-mdec, {C})};Spec = {(AB, ab-spec, {A, B})};*Coup* = {(abc-dec, {(AB.out, CS.in), (CS.out, AB.in)}), (cs-mdec, {(CS.in, C.in), (C.out, CS.out)})};*Rules* = {(ab-spec, {A, B}), (cs-mdec, {selection of kernel-models, number of entities})}.

After the pruning through an algorithm, the PES tree of Figure 2b) is defined as follows:E = {ABC, CS, A, C};A = {A-dec, cs-mdec};Asp = {(ABC, abc-dec, {A, CS}), (CS, cs-mdec, {C}, {broadcast-models, 10})};Coup = {(abc-dec, {(A.out, CS.in), (CS.out, A.in)}), (cs-mdec, {(CS.in, C.in), (C.out, CS.out)})}.

### 3.2. Model Base

The model base represents the behavioral characteristics of the system and describes procedural characteristics. It is composed of models that provide a dynamic and symbolic means of expression. This base store executes software modules of dynamic system components referenced by SES leaf nodes (Figure 2c) [50]. The models stored in the base can be mutually coupled by a given coupling relationship, and then can be constructed into a simulation model (Figure 2d). The DEVS model, which is a representative formalism for discrete event modeling, measures the behavior of a system for discrete events in a continuous time. An atomic DEVS model is formally defined by the following structure
(3)$$\langle X,S,Y,\delta_{int},\delta_{ext},\lambda ,t_{a}\rangle .$$

In Equation (3), *X* is an external input set, *S* is a sequential state set, *Y* is an external output set, $\delta_{int}$ is an internal transition function, $\delta_{ext}$ is an external transition function, λ is an output function, and $t_{a}$ is a time advance function.

Therefore, the SES/MB framework is able to synthesize and execute a structural and hierarchical simulation model by trees of entity structures and atomic DEVS models in bases.

## 4. WSN-SES/MB Framework

We propose the novel SES/MB framework for large-scale WSNs. The WSN-SES/MB uses our transformation process to generate the executable simulation models for the WSNs through the structural representation, the environment configuration, and the behavioral models. In this section, our framework introduces them and shows how they are synthesized into a simulation model.

### 4.1. Overview

The WSN-SES/MB framework automatically synthesizes various simulation models through the structural representation and the network’s environment configuration of a sensor network using our transformation process to save synthesizing time and cost. Our framework’s transformation process is as follows
(4)SES→Proposed Pruning→PES→Composition Tree+Environment Parameters+ MB→Simulation Model.

In Equation (4), our proposed framework generates a simulation model using the entity structure, the configuration, the network environment parameters, and the atomic DEVS models through its transformation process. In the proposed framework, our pruning algorithm generates a PES tree from an SES representing the hierarchical structure of a sensor network. This PES contains a composition tree for the network structure and the environment parameters. Thus, the WSN-SES/MB synthesizes the composition tree and the parameters of the PES, and the atomic models in a model base, to generate a single simulation model.

The WSN-SES/MB framework uses five phases for synthesizing the structures and the models as follows:SES base: This base contains the entity structures of the WSNs. The entity structures show the components of the WSNs hierarchically in various trees;Proposed Pruning: Our pruning algorithm selects entities from an SES tree for generating a PES and configures a sensor network’s environment parameters;PES base: This base includes pruned entity structures of the WSNs. Each entity structure contains the environment parameters;Model base: Atomic DEVS models defined in this base are entities corresponding to the leaf of the SES and PES trees. These atomic models represent the behavioral and procedural characteristics of the sensor network. The models are synthesized to generate the final simulation model;Simulation Model: This simulation model is the result of our framework, which is finally created by the structural representation, the network’s parameters, and the behavioral models.

Therefore, our WSN-SES/MB framework achieves time and cost efficiency while synthesizing the composition tree, the environment parameters and models using our transformation process. The proposed framework increases the diversity of synthesized simulation models and facilitates modeling simulation for WSNs.

### 4.2. Entity Structures

This section introduces an SES base, the proposed pruning algorithm, and a PES base for the WSN-SES/MB framework. Figure 3 shows exemplary entity structures of a sensor network in two bases with the proposed pruning algorithm. In terms of the validation of equivalence between the SES and the PES, when the SES is designed, the model developer already takes the PES into account. Therefore, if the SES is well designed and developed, the PES is inevitably equivalent with a specialized SES. However, for the verification of the PES models, we should monitor the PES’s model execution logs and results to see if it is working as we expected.

#### 4.2.1. SES Base

The SES base (Figure 3a) contains SES trees of the sensor network. An EF-WSN of an SES tree is a top-level entity and consists of WSN and EF, entities including the elements of the sensor network. The WSN entity has the tree leaves of BS and SENSORS. The SENSORS define an entity *sensors-mdec* as representing multiple SENSOR entities based on isomorphism. The EF entity includes GENR and TRAND entities. The GENR randomly generates event data within the sensor field, and the TRANSD collects and analyzes the data generated by the GENR. There is a decomposition between all entities, and the decomposition owns coupling relations. This entity structure in Figure 3a is defined as follows:*E* = {EF-WSN, WSN, EF, BS, SENSORS, SENSOR, GENR, TRANSD};*A* = {ef-wsn-dec, wsn-dec, ef-dec, sensor-mdec};*S* = { };*Asp* = {(EF-WSN, ef-wsn-dec, {WSN, EF}), (WSN, wsn-dec, {BS, SENSORS}), (SENSORS, sensors-mdec, {SENSOR}), (EF, ef-dec, {GENR, TRANSD})};Spec = {};*Coup* = {(ef-wsn-dec, {(EF.out, WSN.in), (WSN.out, ef.in)}), (wsn-mdec, {(WSN.in, SENSORS.in), (BS.bs_out, WSN.out), (SENSORS.out, BS.packet_in}), sensors-mdec, {(SENSORS.in, SENSOR.event_in), (SENSOR.packet_out, SENSOR.packet_in), (SENSOR.sensor_out, SENSORS.out)}, ef-dec, {(EF.in, TRANSD.solved), (GENR.out, EF.out), (TRANSD.out, EF.result), (TRANSD.out, GENR.stop)}};*Rules* = {(sensors-mdec, {sub-model selection of kernel-models, number of the entity, {sensor filed size, BS location}})}.

#### 4.2.2. Proposed Pruning Algorithm

In WSN-SES/MB, our pruning algorithm (Figure 3b) converts into one PES according to the *Rules* of the SES tree.

Algorithm 1 shows our pruning algorithm from an SES tree to a PES for a sensor network. In the pruning algorithm, lines 10 to 13 are newly added for our WSN-SES/MB. Here, the SES tree represent the entity structure of a sensor network. The algorithm inputs the SES tree and searches specialization and multiple-decomposition nodes for the pruning. If a node is a specialization entity, a user can select only one of its child entities. On the other hand, if a node is a multiple-decomposition entity, the user determines a sub-model of DEVS kernel-models and the number of sub-models. If the sub-model is a broadcast-model, a user inputs a sensor field size and a BS location ([10,49] suggests that the broadcast-model is suitable for WSNs).
**Algorithm 1** Pruning of WSN-SES/MB1:   **while**
*node* of all nodes of *SES*2:       **if**
*node* == *specialization*
**then**3:             Select an *entity* among all child *entities;*4:       **end if**5:6:       **if**
*node* == *multiple-decomposition*
**then**
7:             Select a *sub-model* of DEVS *kernel-models;*8:             Input number of members9: 10:            **if**
*sub-model* == ‘*broadcast-model*’ **then**11:                  Input sensorFieldSize(X, Y);12:                  Input bsLocation(X, Y);13:            **end if**14:         **end if**15:     **end while**

#### 4.2.3. PES Base

The PES base contains entity structures that are pruned using the proposed algorithm. In Figure 3c), a PES in the base is a tree pruned from the SES. The tree’s parameters are determined according to the pruning algorithm. The following parameters may be configurated as an example:DEVS Kernel’s Sub-model: broadcast-model;Number of members: 200;Sensor field size: 500 × 500 m^2^;BS Location: 250, 250.

The entity structure in Figure 4c) is defined as follows:*E* = {EF-WSN, WSN, EF, BS, SENSORS, SENSOR, GENR, TRANSD};*A* = {ef-wsn-dec, wsn-dec, ef-dec, sensor-mdec};*Asp* = {(EF-WSN, ef-wsn-dec, {WSN, EF}), (WSN, wsn-dec, {BS, SENSORS}), (SENSORS, sensors-mdec, {CS}, {*broadcast-model*, 200, {(500, 500), (250, 250)}}), (EF, ef-dec, {GENR, TRANSD})};*Coup* = {(ef-wsn-dec, {(EF.out, WSN.in), (WSN.out, ef.in)}), (wsn-mdec, {(WSN.in, SENSORS.in), (BS.bs_out, WSN.out), (SENSORS.out, BS.packet_in}), sensors-mdec, {(SENSORS.in, SENSOR.event_in), (SENSOR.packet_out, SENSOR.packet_in), (SENSOR.sensor_out, SENSORS.out)}, ef-dec, {(EF.in, TRANSD.solved), (GENR.out, EF.out), (TRANSD.out, EF.result), (TRANSD.out, GENR.stop)}}.

Such a PES definition exactly represents the structure and the environment parameters of one sensor network. A fully configured and executable simulation model can be generated from the PES with the atomic DEVS models of the model base.

### 4.3. Model Base

This model base is composed of atomic models that represent the behavioral characteristics of a sensor network and provide a dynamic and symbolic means of expression. Each model of this base is a leaf entity node of am SES or PES tree, expressed as an atomic DEVS model.

Figure 4 shows diagrams of atomic models stored in the model base and pseudo-codes of their main functions. These diagrams and pseudo-codes are executed as follows:SENSOR model (Figure 4a): A sensor model has two roles in a field: (1) a source model and (2) an intermediate model. (1) The source model receives events from an input port *event_in* and generates a report (state transition: *passive*→*sensing*). The report is forward to the next SENSOR through an output port *report_out* (state transition: *sensing*→*forwarding*). (2) The intermediate model receives a report from its source or previous model through an input port *report_in* (state transition: *passive*→*sensing*). This model analyzes the report data and transmits it to the next SENSOR through the output port *report_out* (state transition: *sensing*→*forwarding*). If the next forwarding node is a BS, this model forwards the report through an output port *sn_out*;BS model (Figure 4b): This model receives the report sent from SENSOR through an input port *report_in* (state transition: *passive*→*receiving*). After the model analyzes the report’s data (state transition: *receiving*→*analyzing*), it notices the report result through an output port *out*;GENR model (Figure 4c): This model randomly generates events in the *active* state. The event message is delivered to SENSOR and TRANSD through an output port out. When a message is received form TRANSD through an input port *stop*, the event is no longer generated (state transition: *active*→*passive*);TRANSD model (Figure 4d): This model collects event data generated from GENR through an input port *ariv* and result information from the report received from BS through a port solved. The model evaluates the simulation model through the collected analysis results. When the simulation observation time is over, a stop message is output through a port *out* and this message is delivered to GENR (state transition: *active*→*stop*).

### 4.4. Simulation Model

The WSN-SES/MB generates a simulation model for the sensor network by automatically synthesizing the PES tree, the environment parameters, and the atomic DEVS models. The structure of this generated model is constructed according to the structure of the PES tree, and the environment configuration in the network is built according to the parameters. In addition, the model’s behavior depends on the atomic models.

Figure 5 shows a target DEVS model for a sensor network with modular and hierarchical features created based on the WSN-SES/MB. The number of the SENSOR is generated according to an environment parameter set during pruning. Moreover, the location of the BS is determined according to the parameters (X and Y coordinates). After starting the simulation model, the GENR model runs first. The GENR checks a field size of other configurated parameters and generates an event within the field. The SENSROS, which is a sub-model of the DEVS kernel, receives the event and selects a SENSOR model based on the location of the event. The source SENSOR model generates a report through the event and forwards it to the next SENSOR model. The next model receives the report and forwards it. When the BS receives the report, the BS analyzes the report and informs the TRANSD of the report’s results.

## 5. Simulation Results

We performed a simulation experiment to evaluate the WSN-SES/MB and compare it to the DEVS-C++ simulator. This existing simulator should additionally work the implementation of coupled DEVS models and the configuration of the environment parameters to manually synthesize the models and the parameters.

The sensor field was proportionally determined according to the number of sensors in consideration of the density of nodes as shown in Table 2. The sensors forwarded reports to the BS via multiple hops [9,52]. The transmission range of the sensor nodes is 100 m [9]. The location of the BS is freely determined according to parameters. We set the behavioral time of each state for atomic DEVS models based on [9]. In the simulation experiment, we randomly generated 100 events. There was no packet loss in the experiment. The observation time of the simulation is 200, where an event is generated at every time of 2.

Figure 6 illustrates the execution time of our WSN-SES/MB framework and the DEVS-C++ according to the number of sensor models. The WSN-SES/MB synthesizes the entity structure and the atomic DEVS models; the DEVS-C++ synthesizes atomic and coupled DEVS models. Although our framework consumes an almost constant execution time synthesizing between 50 and 200 models, it improves the time by up to 19% when synthesizing between 500 and 1000 models. On the other hand, the DEVS-C++ increases the time proportionally between 50 and 200 models, and it increases the time rapidly between 500 and 1000 models. In addition, apart from the execution time of the DEVS-C++, additional hardcoding and the code build times are also consumed. In modeling and simulation, to obtain the optimization model, these technologies should alter model parameter values and restart simulations more than dozens of times [53]. Therefore, the proposed WSN-SES/MB framework uses our transformation process to save time in running a synthesis of DEVS models.

Figure 7 shows the average ratio of CPU utilization measured according to the number of sensor models while synthesizing the DEVS models. The WSN-SES/MB utilized an average of 7.6% of the CPU, and the DEVS-C++ worked an average of 11.6% of the CPU resources. Overall, our proposed framework saved about 4% of the CPU resources as compared to the DEVS-C++.

Figure 8 illustrates the average memory utilization measured according to the number of the sensors while synthesizing the models. The WSN-SES/MB used an average of 36.04 megabytes of the random-access memory (RAM), the DEVS-C++ operated average of 37.32 megabytes of the RAM. Overall, our framework minutely saved the memory use when compared to the DEVS-C++.

Thus, the proposed framework saved both execution time and CPU resource use while maintaining the memory use as compared to the DEVS-C++ when the models were synthesized.

Figure 9 shows the average turnaround measurement time according to the number of sensors and the BS locations (i.e., the top center, the middle, the bottom center). Overall, it was confirmed that the simulation models operated normally in various environments. As shown in this figure, the simulation models with the BS in the middle spend a shorter average turnaround time than other models. This was because the intermediate SENSOR models forwarded reports to the BS within a short number of hops. Thus, our framework worked normally, despite the different number of sensor nodes, field sizes, and BS locations based on the proposed transformation process.

Thus, the WSN-SES/MB achieves cost and time savings using our transformation process through the simulation results. Using the transformation technology, the proposed framework increases the diversity of synthesized simulation models and facilitates the modeling and simulation of large-scale WSNs.

## 6. Conclusions and Future Work

In large-scale WSNs, it is important to predict situations through various simulations before sensor nodes are distributed to the field, because it uses a lot of resources to change their configuration once installed. Before the deployment of a WSN, some techniques, such as modeling and simulation, should be used to predict the behavior and performance of the network. In addition, it is necessary to reduce unneeded time and costs when evaluating the various performances of simulation models.

In this paper, we propose the WSN-SES/MB framework to effectively save cost and time during automatic model synthesis through our transformation process while modeling and simulating. Our proposed scheme was able to achieve:Application of the SES/MB to synthesize the structure and models and automate the complicated synthesizing process of WSN modeling construction. Our WSN-SEN/MB framework automatically synthesizes the structure and simulation models through the transformation process;Reduced cost and time when synthesizing various sensor networks by proposing a novel transformation process. In the propsoed framework, our transformation process effectively reduces the cost and time of automatic synthesis through the proposed pruning algorithm;Efficiency in the WSN-SES/MB framework with our pruning algorithm. The proposed pruning algorithm configures environment parameters of the sensor network, such as the number of sensors, the field size, and the BS location. These parameters are used during the synthesis of the DEVS model;Increased diversity of synthesized simulation models. The WSN-SES/MB framework generates simulation models using the entity structures within the parameters and the atomic DEVS models;The modeling and simulation of large-scale WSNs. The proposed framework enables the construction of large-scale WSNs through various environment parameters.

Therefore, the WSN-SES/MB framework achieves cost and time savings using our transformation process as compared to DEVS-C++ technology. In addition, our proposed framework increases the diversity of synthesized simulation models and facilitates the modeling and simulation of large-scale WSNs. In future work, we will study synthesis techniques by applying routing and security protocols to WSNs.

## Figures and Tables

**Figure 1 sensors-21-00430-f001:**
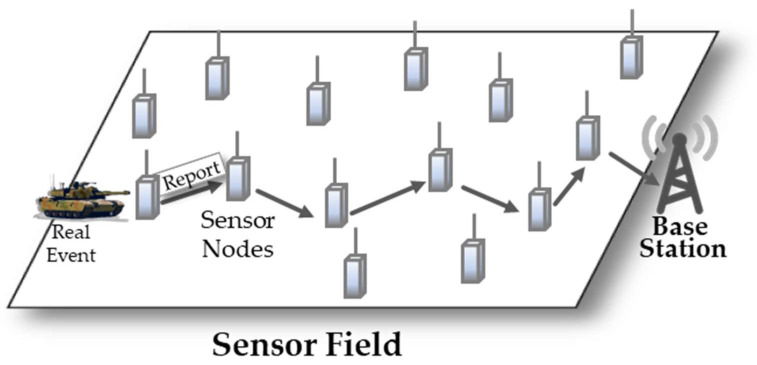
Wireless sensor network.

**Figure 2 sensors-21-00430-f002:**
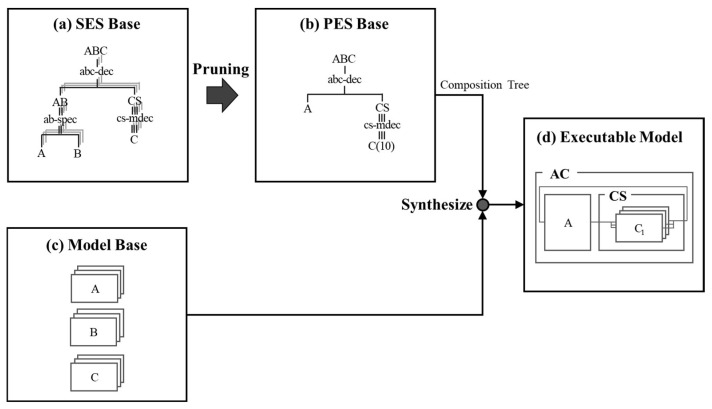
Transformation Process in SES/MB framework.

**Figure 3 sensors-21-00430-f003:**
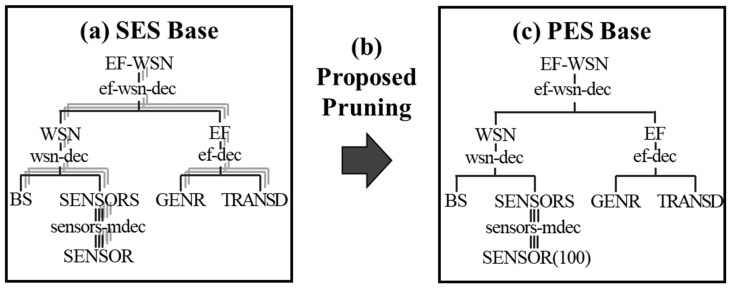
SES and PES for the sensor network.

**Figure 4 sensors-21-00430-f004:**
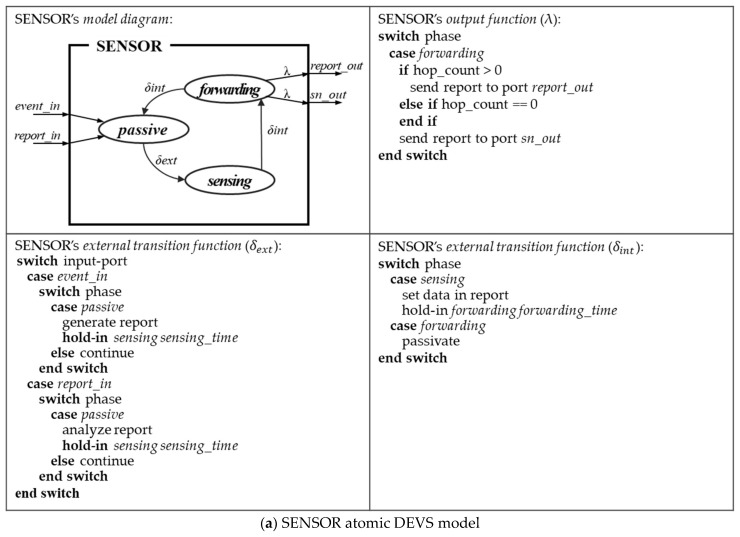
Atomic DEVS models in MB.

**Figure 5 sensors-21-00430-f005:**
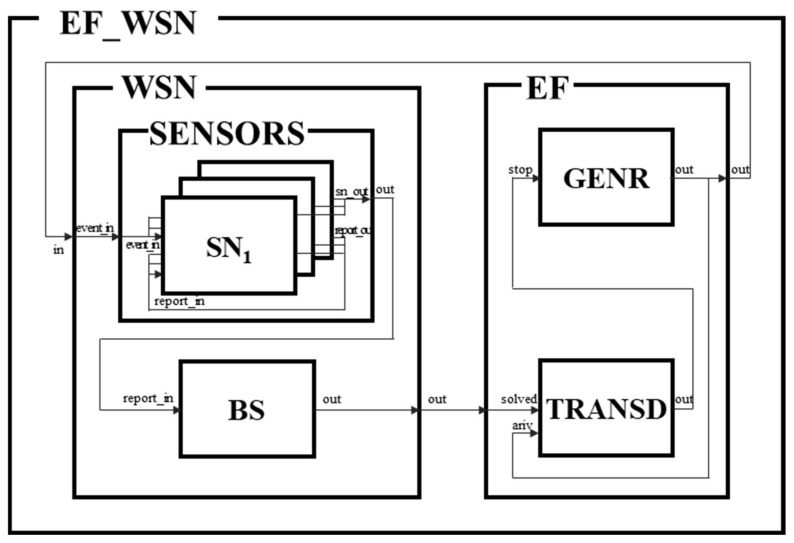
Executable Simulation Model of a WSN.

**Figure 6 sensors-21-00430-f006:**
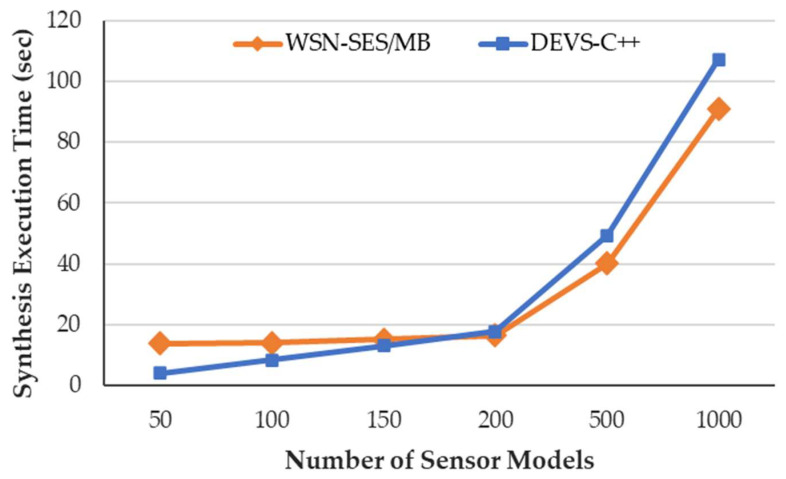
Synthesis execution time versus the number of sensor models.

**Figure 7 sensors-21-00430-f007:**
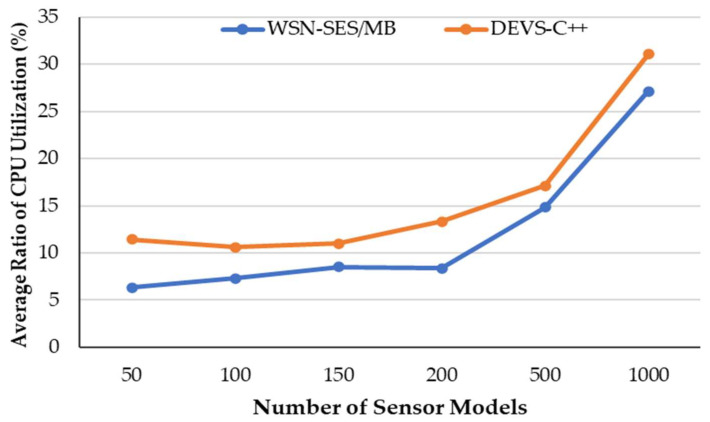
Ratio of CPU utilization versus number of sensor models while synthesizing the models.

**Figure 8 sensors-21-00430-f008:**
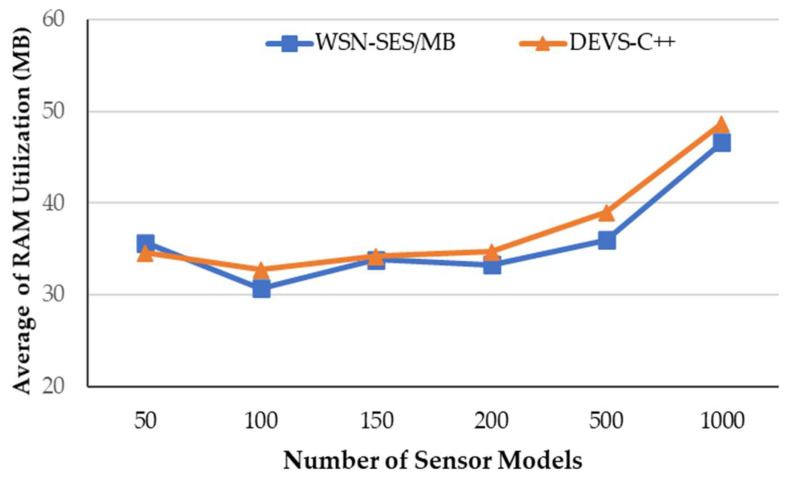
Average of RAM utilization versus number of models while synthesizing the models.

**Figure 9 sensors-21-00430-f009:**
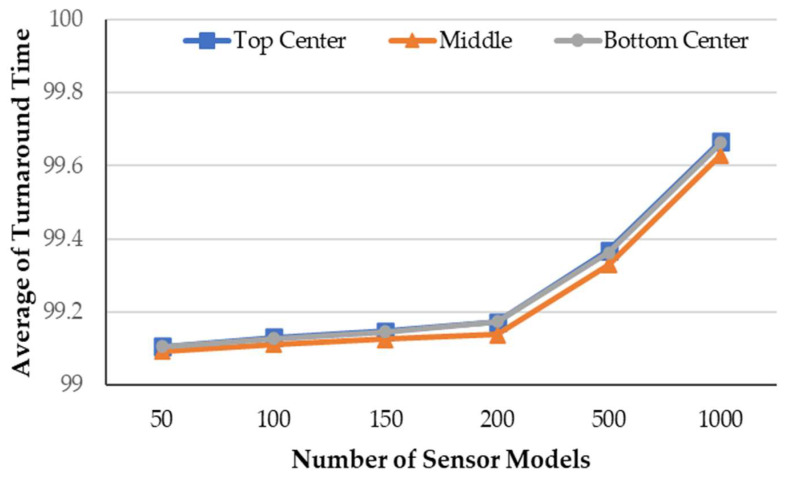
Average of Turnaround Time versus number of sensor nodes according to the BS locations.

**Table 1 sensors-21-00430-t001:** Wireless sensor network (WSN) simulator comparison.

No.	Simulator	Programming Language	Type	Characteristic	Automatic Model Synthesis	Limitations
1	Network Simulator-3 (NS-3)	C++	Discreate-Event Simulation	General Simulator	No	Model Diversity and Reuse, Insufficient results and analysis methods
2	Objective Modular Network Testbed in C++ (OMNET++)	C++	Discreate-Event Simulation	General Simulator	No	Model Diversity and Reuse, Insufficient results and analysis methods
3	TinyOS simulator (TOSSIM)	nesC	Discreate-Event Simulation	Specifically designed for WSNs	No	Only for TinyOS Code
4	probabilistic wireless network simulator (Prowler)	nesC	Discreate-Event Simulation	Specifically designed for WSNs	No	Probabilistic traffic, Insufficient results and analysis methods
5	simple NEST application simulator (Siesta)	nesC	Dynami Simulation	Specifically designed for WSNs	No	Only for TinyOS Code
6	Ashut	nesC	Discreate-Event Simulation	Specifically designed for WSNs	No	Only for TinyOS Code
7	routing modeling application simulation environment (Rmase)	nesC	Discreate-Event Simulation	Specifically designed for WSNs	No	Only for TinyOS Code
8	Sensor, environment and network simulator (SENS)	C++	Discreate-Event Simulation	Specifically designed for WSNs	No	Model Diversity and Reuse
9	Fine-Grained Sensor Network Simulator (ATEMU)	nesC	Discreate-Event Simulation	Specifically designed for WSNs	No	Only for TinyOS Code
10	CupCarbon	Java	Discreate-Event Simulation	Specifically designed for WSNs	No	Model Diversity and Reuse, Insufficient results and analysis methods
11	DEVS-C++	C++	Discreate-Event Simulation	Specifically designed for WSNs	No	Manual synthesis

**Table 2 sensors-21-00430-t002:** Experimental environment of the field according to the number of sensors.

Number of Sensors	50	100	150	200	500	1000
Sensor of Field (m^2^)	200 × 200	300 × 300	400 × 400	500 × 500	1000 × 1000	2000 × 2000

## Data Availability

Data of this research is available upon request via corresponding author.

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
