# Peer review of "WSN-SES/MB: System Entity Structure and Model Base Framework for Large-Scale Wireless Sensor Networks"

_sensors, 2021, doi:10.3390/s21020430_

Round 1
Reviewer 1 Report
Overall, this is a nice and interesting work on simulation framework for WSNs. Still, there are a number of major concerns that I recommend the authors to address before publication: 1) In the introduction part, the authors should better motivate the practical background of this research. There have been many modelling and simulation frameworks for WSNs, What is the novelty of this framework. 2) The list of contributions in Sec. I should be expanded and enriched so as to better highlight the technical challenges tackled by the authors. 3) When introducing the basic concept and application of WSNs, the following references may be considered for the sake of a more complete introduction to the context: “Modeling and Optimizing the Cascading Robustness of Multisink Wireless Sensor Networks, IEEE Trans on Reliability”, “Cascading failures in wireless sensor networks with load redistribution of links and nodes, Ad Hoc Networks”, “Modeling Cascading Failures for Wireless Sensor Networks With Node and Link Capacity-IEEE Trans on Vehicular Technology”. 4) In the related work, I highly recommend the authors to add a comparison table to illustrate the limitations of the existing frameworks. By this way, the contributions and novelty of the proposed framework can be better highlighted. 5) In the simulation part, I suggest the authors to add some comparison experiments with other widely-used simulation platforms. 6) More simulation details should be given, such as the wireless interference model, energy consumption model. 7) Another of my concern is that can the proposed platform support the mobile sensor networks? 8) The authors should carefully proofread the paper. Some typos can be found in the paper, such as in the line 445, an extra “s” can be found after the sentence “routing and security protocols to WSNs”;Author Response
Dear the reviewer of the Sensors,
We would like to acknowledge the referees for their professional and excellent recommendations for our submitted article. We have revised our manuscript with regards to the comments posed by the referees. Please find our response below. We hope that the revision meets both of the referees' and your approval. Please check the attached file.
Thank you very much.
Best regards,
Hyung Jong Kim.

Reviewer 2 Report
This paper presents a SES/MB framework for WSN, and has the following drawbacks.
- The contribution of the paper seems rather minor. According to Figure 6, the execution time is only about tens of seconds. Is it really meaningful to reduce such short times additionally? In addition, for most of the numbers in the same figure, the manual synthesis is better. Furthermore, the SES/MB framework is already available [42–46]. This paper merely adopts it to the WSN problems, limiting the novelty.
- The evaluation has not been conducted rigorously. As introduced at the early part of the paper, there exist many prior simulators in the literature. However, the proposed framework is not compared with most of them. It is currently hard to say that the proposed work is effective.
- In Figure 5, why are the BS, GENR, TRANSD drawn so big?
Author Response
Dear the reviewer of the Sensors,
We would like to acknowledge the referees for their professional and excellent recommendations for our submitted article. We have revised our manuscript with regards to the comments posed by the referees. Please find our response below. We hope that the revision meets both of the referees' and your approval. Please check the attached file.
Thank you very much.
Best regards,
Hyung Jong Kim.

Reviewer 3 Report
The authors present a new framework to simplify the simulation of wireless sensor networks. The paper is well written and well structured. The different sections reflect the difficulty of the proposal. English is adequate. References should be checked because most of them cite three or four articles. The authors should carry out more tests and present an example of the proposal. Some additional questions are raised below.
Q1. Line 27. Change the references [1,2] to more current ones. For example:
Landaluce, H., Arjona, L., Perallos, A., Falcone, F., Angulo, I., & Muralter, F. (2020). A Review of IoT Sensing Applications and Challenges Using RFID and Wireless Sensor Networks. Sensors, 20 (9), 2495.
Q2. Line 35. Cite only one paper.
Q3. Line 40. Change the reference [5] for a more current paper.
Q4. Line 57. 4 papers are cited. You have to add paragraphs to cite the articles one by one.
Q5. Line 158. Text about Fig 2 must appear before the figure.
Q6. Section 4.1 How do you check that the pruning process is equivalent to the original process?
Q7. Line 336. Text about Fig 4 must appear before the figure.
Q8. Table 1. More tests must be added: 500, 1000 and 2000 sensors minimum.
Q9. What is the energy model used in the simulations? How are user metrics obtained in sensor networks?
Q10. Any image of the application?
Q11. How does the user interact with the framework? Is there an example of the pseudocode that a user must enter?
Q12. The results in Fig. 6 How have they been obtained?
Author Response

(The authors gave the same response as above.)

Reviewer 4 Report
Specific comments and remarks:
- In this paper, how are the advantages of the proposed model defined in the abstract proven? The authors have to clearly represent the advantages and disadvantages of their model;
- The authors have to perform a comparison with more similar models published in the literature. In this way, it will be easier to highlight the advantages of the proposed model;
- Since this work lacks experimental results from the actual work of the proposed model, the authors have to explain in more detail how to save "cost and time" by offering a specific framework.
Author Response

(The authors gave the same response as above.)

Round 2
Reviewer 1 Report
I think the manuscript is qualified for acceptance.
Reviewer 2 Report
The authors have improved the manuscript according to the concerns raised. The reviewer appreciates the efforts of the authors.
Reviewer 3 Report
From my point of view the authors have made the suggested changes appropriately.